# Amoxicillin-Induced Neurotoxicity: Contribution of a Healthcare Data Warehouse to the Determination of a Toxic Concentration Threshold

**DOI:** 10.3390/antibiotics12040680

**Published:** 2023-03-30

**Authors:** Sébastien Lalanne, Guillaume Bouzillé, Camille Tron, Matthieu Revest, Elisabeth Polard, Eric Bellissant, Marie-Clémence Verdier, Florian Lemaitre

**Affiliations:** 1Department of Pharmacology, CHU Rennes, Inserm, EHESP, Irset (Institut de Recherche Ensanté, Environnement et Travail) UMR_S 1085, University of Rennes, 35000 Rennes, France; 2CHU Rennes, Inserm, LTSI—UMR 1099, University of Rennes, 35000 Rennes, France; 3CHU Rennes, Infectious Diseases and Intensive Care Unit, 2 Rue Henri Le Guilloux, University of Rennes, 35033 Rennes, France

**Keywords:** amoxicillin, neurotoxicity, concentrations

## Abstract

Background: Amoxicillin (AMX)-induced neurotoxicity is well described and may be associated with AMX overexposure. No neurotoxic concentration threshold has been determined thus far. A better knowledge of maximum tolerable AMX concentrations is of importance to improve the safety of high doses of AMX. Methods: We conducted a retrospective study using the local hospital data warehouse EhOP^®^ to generate a specific query related to AMX neurotoxicity symptomatology. All patient medical reports containing a mention of neurotoxicity clinical symptoms coupled with AMX plasma concentration measurements were explored. Patients were classified into two groups according to the imputability of AMX in the onset of their neurotoxicity, on the basis of chronological and semiological criteria. A receiver-operating characteristic curve was performed to identify an AMX neurotoxic steady-state concentration (Css) threshold. Results: The query identified 101 patients among 2054 patients benefiting from AMX TDM. Patients received a median daily dose of 9 g AMX, with a median creatinine clearance of 51 mL/min. A total of 17 of the 101 patients exhibited neurotoxicity attributed to AMX. The mean Css was higher for patients with neurotoxicity attributed to AMX (118 ± 62 mg/L) than those without 74 ± 48 mg/L (*p* = 0.002). A threshold AMX concentration of 109.7 mg/L predicted the occurrence of neurotoxicity. Conclusions: This study identified, for the first time, an AMX Css threshold of 109.7 mg/L associated with an excess risk of neurotoxicity. This approach needs to be confirmed by a prospective study with systematic neurological evaluation and TDM.

## 1. Introduction

Amoxicillin (AMX) is a widely used beta lactam (BL). High doses of AMX are required for hospitalized patients for the management of severe infections, such as bacteremia or infective endocarditis. BL-induced neurotoxicity has been clinically described since 1945 [1], with an incidence of up to 15% in intensive care units [2]. In addition, BL-induced neurotoxicity presents a non-specific panel of clinical manifestations that do not contribute to the identification of neurotoxicity linked to this class.

The mechanisms of BL-induced neurotoxicity are generally related to the lactam ring, the ability of BLs to cross the blood–brain barrier, and their interaction with the GABA-a receptor [2]. Electroencephalogram abnormalities may help to characterize BL-induced neurotoxicity, such as for cefepime, with generalized periodic discharge and nonconvulsive status epilepticus (NCSE), but are not specific to this drug class [3].

Data supporting AMX neurotoxicity are limited, especially in clinical trials, but have already been documented in pre-clinical models [4,5]. BL therapeutic drug monitoring (TDM) is routinely performed in numerous centers for the management of complex infections and has been the subject of guidelines to mainly reach effective concentration targets. Neurotoxic concentration thresholds have also been identified in prospective or retrospective clinical trials for cefepime and meropenem but not for AMX [6]. TDM is proposed for AMX for patients requiring high doses but target concentrations are determined to improve efficacy [6]. It is assumed that neurotoxicity is associated with high concentrations, but the threshold related to toxic concentrations is not well defined for this antibiotic. Thus, we matched the clinical description and drug concentration data from AMX TDM to evaluate the concentration-dependent toxicity of the drug and identify a maximum tolerable plasma AMX concentration.

## 2. Results

Among the 2054 patients benefiting from AMX TDM in the preliminary extraction, 101 analyzable patients were identified by the “neurotoxicity query”. The median age was 73 [14–91] years and the patients received a median AMX daily dose of 9 g [2–21], and had a median estimated glomerular filtration rate (eGFR) of 51 (22–86) mL/min. The main infections were infective endocarditis (35%), osteo-articular infections (20%), and blood (16%) (*n* = 101). Seventeen patients presented neurotoxicity attributed to AMX (cases) and 84 did not (controls).

The median daily dose was 9 (6–12) versus 9 (6–11) g (not significant) for cases and controls, respectively. The median eGFR was statistically different, with cases having a lower eGFR (22 (12–56) mL/min) than controls (57 (28–87) mL/min) (*p* = 0.02) (Figure 1). The mean Css was 118 ± 62 mg/L for cases and 74 ± 48 mg/L for controls (*p* = 0.002) (Figure 2). After the multivariable analysis, the only parameter that significantly influenced the occurrence of neurotoxicity was AMX Css (*p* = 0.006). ROC curve analysis showed a maximal Youden Index for AMX concentration of 109.7 mg/L. This threshold provided a sensitivity of 59%, (95% confidence interval [35–82]) and high specificity (86%), (95% confidence interval [78–93]) to differentiate patients with neurotoxicity attributed to AMX or not. The calculated area under the ROC curve vas 0.73, 95%CI (0.58–0.96) (Figure 3). ROC curve obtained by including into the model clinically relevant variables (age, sex, weight, AMX daily dose, infection type, eGFR) although not identified as statistically significant in the multivariate analysis identified the same threshold value (Figure 4).

Exceeding the 109.7 mg/L threshold was found to be associated with a 7.5-fold occurrence of AMX-induced neurotoxicity (95% confidence interval [2.38–23.44]).

## 3. Discussion

There is evidence to document a link between drug exposure and the occurrence of BL-induced neurotoxicity. For piperacillin, meropenem, and flucloxacillin, a Cmin > 361.4, 64.2, and 125.1 mg/L, respectively, were shown to be associated with a 50% risk of developing a neurotoxicity event in a retrospective study [7], whereas different thresholds were documented for cefepime (Cmin > 20 to Css > 63 mg/L) [8,9]. Mechanisms supporting BL-related neurological adverse events have been previously described in several studies and reviews and reported clinical manifestations were confusion, vigilance disorders, shaking or myoclonus, a decreased level of consciousness, and epilepsy [2,10,11]. High-dose therapy, renal impairment, preexisting or underlying neurological abnormalities, and age are the main identified risk factors for beta lactam-induced neurotoxicity [10].

Clinical symptoms related to AMX or ampicillin have already been described in clinical studies, such as myoclonia [12], delirium onset [13] (with a 95% confidence interval odds ratio of 2.13–9.45 for the ampicillin sulbactam combination), convulsions [14], and behavioral changes [15]. These data are supported by animal models with evidence for epileptogenicity and a depression-like response following direct intraventricular injections [4]. Furthermore, oral AMX administration (25 and 50 mg/kg) significantly reduced latency to the first induced seizure in a pentylenetetrazole rat model of epilepsy [5]. To date, no concentration–neurotoxicity threshold has been documented for AMX. Marti et al. conducted a prospective trial in 156 critically ill patients and found no neurotoxicity in a cohort of 156 patients but these patients were treated with a low daily dose (4 g) of amoxicillin administered by discontinuous infusion [16].

Herein, we report a threshold concentration of 109.7 mg/L to identify patients with AMX-related neurotoxicity. The similar thresholds obtained using two approaches for ROC curve analysis strengthen this result. The adjusted ROC curve strategy allows us to exclude potential confounding factors as described by Inácio et al. [17]. The use of a data warehouse allowed us to identify a very low-frequency adverse effect (<1% of patients) within a population with documented risk factors for AMX neurotoxicity: high antibiotic daily dose (9 g/24 h), moderate renal failure, and age >65 years. This threshold is consistent with the current upper concentration threshold for AMX TDM of 80 mg/L recommended for intensive care unit patients [6]. However, this upper bound did not rely on toxicity evidence but rather on the absence of a benefit for the patient when the concentration was above eight times the minimal inhibitory concentration breakpoint for AMX. One major limitation of our study was that there was no systematic assessment of the neurological status of the patients. Our query only explored neurological disorders reported in the patient’s medical record and for whom an AMX TDM was prescribed. In addition, it was difficult to estimate the burden of the infection and/or the involved pathogen in the occurrence of neurotoxicity. In certain cases, deterioration of the renal clearance of drugs is itself the consequence of an infectious process that leads to antimicrobial overexposure (e.g., septic renal embolism in infective endocarditis). Moreover, neurological secondary localization (septic embolisms) can occur in specific infections. Finally, a limitation of the study is that the analysis of the initial 2054 medical records is not permitted by the data warehouse, and we only performed an exhaustive analysis of the 101 medical records extracted by the neurotoxic query. However, the methodological approach we used for establishing AMX imputability (a blind assessment by two pharmacologists and an independent pharmacologist adjudicator in case of discrepancies between the two first experts) strengthens the confidence in our results. The method is similar to that used in pharmacovigilance studies and its validity has been largely acknowledged. Another option would have been to use the cumulative area under the curve (AUC) rather than the Css, as it may better correlate with the occurrence of neurotoxicity by taking into account the time of exposure, as suggested for cefepime-induced neurotoxicity [10], but, again, the patients of our center treated with high doses of AMX received almost exclusively continuous infusions and, thus, the measured Css correlates directly with the AUC.

## 4. Materials and Methods

This was a monocenter retrospective study. The data for patients who received AMX TDM [18] between 2009 and 2018 were primarily extracted from the local data warehouse EhOP^®^. EhOP^®^ provides anonymized medical information for all computerized data from hospital and outpatient visits to Rennes University Hospital in the form of free text (clinical comments, daily hospitalization reports, drug prescriptions) or biological values. Among this patient population, a specific full-text query (“neurotoxicity query”) was generated with EhOP^®^ to retrieve exhaustive keywords in relation to potential AMX neurotoxicity. All patient medical reports containing the terms confusion, encephalopathy, neurotoxicity, convulsion, and overdosing associated with AMX (international non-proprietary name or brand name) were extracted.

All included patient medical data containing these keywords were blindly reviewed by two pharmacologists. An objective assessment of imputability based on chronological and semiological criteria was performed to classify patients into two groups according to the imputability or not of AMX at the onset of their neurotoxicity. Patients with neurotoxicity attributed to AMX were referred to as “Cases” whereas patients for whom the role of AMX in neurotoxicity was excluded were referred to as “Controls”. In cases of disagreement between the two experts, a third independent pharmacologist was consulted. Demographic, clinical, and biological data were gathered, including maximal steady-state concentrations (Css) and the concomitant eGFR estimated using the CKD-EPI formula [19]. Statistical comparisons were performed using Mann–Whitney tests for continuous variables, expressed as mean ± standard deviation (sd), median [range] or (interquartile range, IQR), with a significance threshold of 0.05. The impact of baseline characteristics (age, sex, weight, AMX daily dose, infection type, eGFR) and AMX Css as potential confounding factors on the onset of AMX-induced neurotoxicity was assessed by multivariable logistic regression analysis. The significance threshold was 0.05. The AMX plasma concentration threshold that differentiated the two groups were identified using receiver operating characteristic (ROC) curve analysis. The threshold was obtained by retaining the maximum value of the Youden Index (sensitivity + specificity-1). The AMX plasma concentrations threshold values were compared using the ROC curve built with and without adjustment of potential confounding factors listed above, in order to exclude the influence of these covariates. All statistical analyses were performed using Prism 5.0^®^ software (GraphPad Software, La Jolla, CA, USA) and SAS statistical software, V9.4 (SAS Institute, Cary, NC, USA).

This study was approved by the Ethics Committee of Rennes (Registration number 22.172).

## 5. Conclusions

In conclusion, we identified a threshold of amoxicillin concentrations in a real-life population associated with an excess risk of neurotoxicity based on a specific search of a local data warehouse. This concentration of 109.7 mg/L is still largely above the current therapeutic concentration range and may provide a better understanding of the concentration-toxicity relationship of AMX. These data need to be confirmed by a prospective trial with systematic TDM, neurological evaluation, and determination of an imputability score.

## Figures and Tables

**Figure 1 antibiotics-12-00680-f001:**
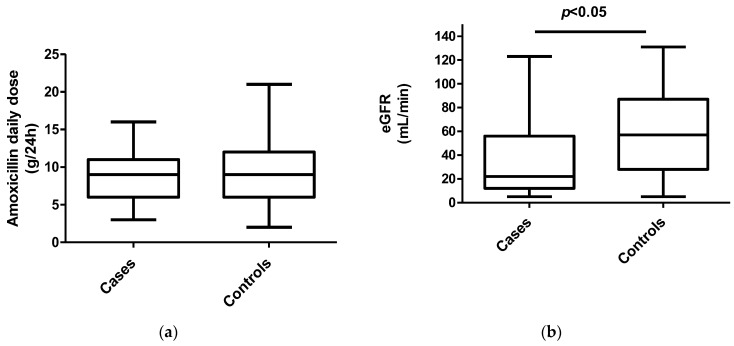
(**a**) Distribution of amoxicillin median daily dose (g), (*n* = 86); (**b**) distribution of estimated glomerular filtration rate (mL/min), (*n* = 101) for patients with neurotoxicity attributed (cases) and non-attributed (controls) to amoxicillin. Horizontal solid lines are the median.

**Figure 2 antibiotics-12-00680-f002:**
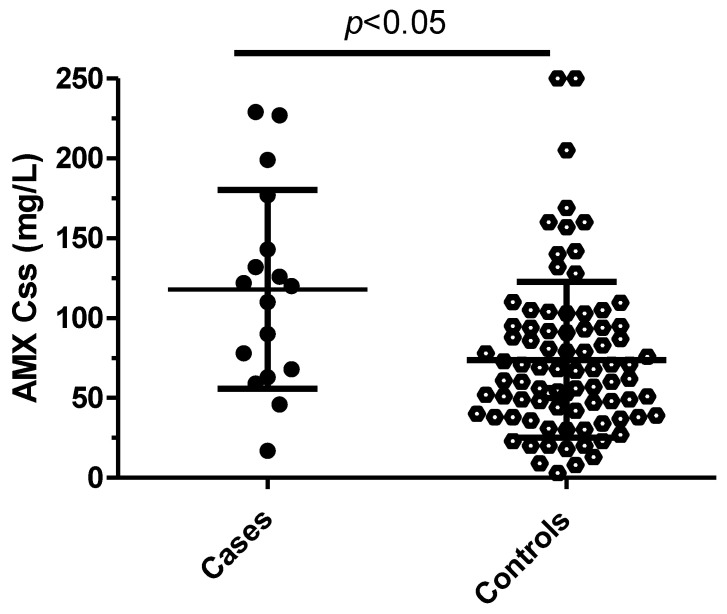
Scatter dot plot of amoxicillin steady-state concentrations (mg/L) for patients with neurotoxicity attributed (cases, *n* = 17) and non-attributed (controls *n* = 84) to amoxicillin. Horizontal solid lines are the mean ± standard deviation (sd) concentrations.

**Figure 3 antibiotics-12-00680-f003:**
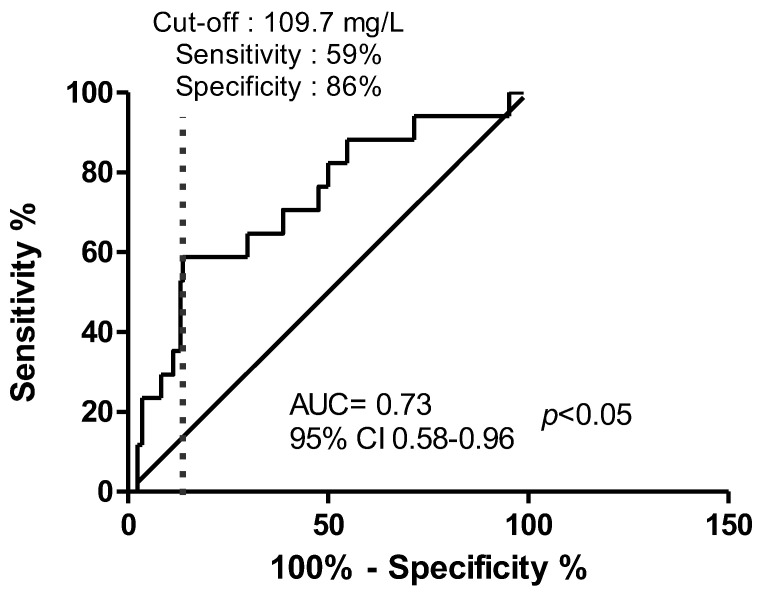
Receiver-operating characteristic curve of amoxicillin plasma concentration in relation to its attributable neurotoxicity or not. AUC: area under the curve; 95% CI, 95%: confidence interval.

**Figure 4 antibiotics-12-00680-f004:**
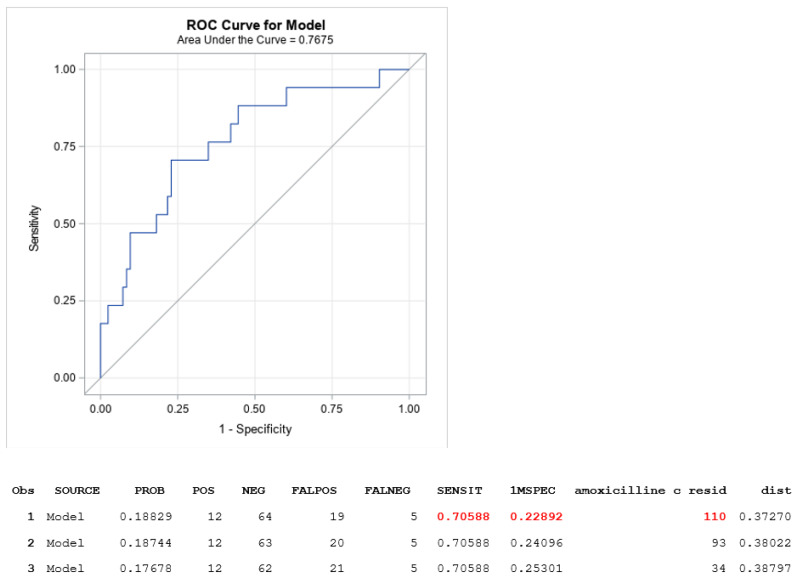
Receiver-operating characteristic curve of amoxicillin plasma concentration in relation to its attributable neurotoxicity or not after adjustement of the potential confounding factors: age, sex, weight, AMX daily dose, infection type, eGFR.

## Data Availability

Not applicable.

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
