# Peer review of "Amoxicillin-Induced Neurotoxicity: Contribution of a Healthcare Data Warehouse to the Determination of a Toxic Concentration Threshold"

_antibiotics, 2023, doi:10.3390/antibiotics12040680_

Round 1

Reviewer 1 Report

Authors used a case-control study using database of clinical data in Rennes University to identify threshold of amoxicillin concentration for neurotoxicity. Because amoxicillin is major beta lactam antibacterial medicine, it is meaningful to identify the threshold. However, this article has not fully answered some of the questions due to inadequate statistical analysis and insufficient description.

First, authors excluded the patients who were not identified by “neurotoxicity query” from 2,054 patients who may receive amoxicillin, which may lead to selection bias. For example, authors used words including “overdosing”, but this exclusion may affect the balance of samples. If the 2,054 patients have received amoxicillin, authors should include all of these patients for analysis.

Second, authors used receiver operating characteristics (ROC) curve to identify the threshold, but authors could not justify why threshold can be identified by ROC curve. In fact, authors mentioned “For piperacillin, meropenem, and flucloxacillin, a Cmin >361.4, 64.2, and 125.1 mg/L, respectively, were shown to be associated with a 50% risk of developing a neurotoxicity event in a retrospective study [4]” (L106), which suggests that these previous studies used “50% risk” for identification of threshold (L107). If these previous studies and this manuscript are comparable, authors should use same method to identify the threshold.

Third, authors used simple ROC without adjustment of potential confounding factors such as age, gender and GFR, but these potential confounding factors may affect the results. Authors should add analysis which adjusts for potential confounding factors.

Forth, authors showed sensitivity and specificity as the result, but the confidence interval were not shown in the manuscript. Authors should add confidence intervals of sensitivity and specificity in the manuscript.

Fifth, authors described some of sentences without citations as follows “The mechanisms of BL-induced neurotoxicity are generally related to the lactam ring, the ability of BLs to cross the blood-brain barrier, and their interaction with the GABA-a receptor.” (L51), “but have already been documented in pre-clinical models.” (L56), “Neurotoxic concentration thresholds have also been identified in prospective or retrospective clinical trials for cefepime and meropenem” (L60), and “TDM is proposed for AMX for patients requiring high doses” (L61). References should be cited for these descriptions.

Finally, authors do not declare the statement of informed consent and approval of institutional ethical review board. As these ethical issues are essential in medical research, authors should add description of these ethical issues.

Minor comment

Abstract: “Css” should be spelled out.

Author Response

Manuscript Reference: antibiotics-1995807 intitled " Amoxicillin-induced neurotoxicity: contribution of a healthcare data
warehouse to the determination of a toxic concentration threshold"

We would like to thank again the referees and the editor for their interesting comment which, we believe, would strengthen the manuscript. We try to answer point by point to each of them.

Reviewer 2 Report

The authors presented an interesting topic. 

It seems that there would be a lot of cofounding factors including GFR, type of infection, etc. How would you consider the effect of those cofounding variables? You may consider a multivariate analysis. 

The definition of neurotoxicity is unclear. How did you know patient's clinical status change was because of amoxicillin? Was that subjective or objective?

Author Response

Manuscript Reference: antibiotics-1995807 intitled " Amoxicillin-induced neurotoxicity: contribution of a healthcare data
warehouse to the determination of a toxic concentration threshold"

Response to the referees

We would like to thank again the referees and the editor for their interesting comment which, we believe, would strengthen the manuscript. We try to answer point by point to each of them.

Reviewer 3 Report

To the Authors

1.       Line 32: I guess that “lower” should be replaced by “higher”…

2.       Please quote and comment the work by Marti et al (Therapeutic drug monitoring and clinical outcomes in severely ill patients receiving amoxicillin: a single-centre prospective cohort study. Int J Antimicrob Agents. 2022 Jun;59(6):106601. doi: 10.1016/j.ijantimicag.2022.106601)

3.       In the abstract, the concentrations of amoxicillin in patients experiencing or not neurotoxicity are given as mean plus standard deviation, whereas in the results section they are given as median [IQR]. Please revise for consistency.

4.       Line 81: please define what is meant for “acceptable sensitivity”. Is there a threshold?

Author Response

(The authors gave the same response as above.)

Round 2

Reviewer 1 Report

Authors revised the manuscript, but this article has not fully answered some of the questions due to inadequate statistical analysis and insufficient description as mentioned in the previous review.

First, as mentioned in the previous review, authors excluded the patients who were not identified by “neurotoxicity query” from 2,054 patients who may receive amoxicillin, which may lead to selection bias. Authors claimed “We assume this bias because the initial query allows us to focus on an event with presumed low frequency. The exhaustive analysis of the 2054 medical records is not permitted by the data warehouse and the analysis of each medical record can only be performed after obtaining the query "neurotoxicity".” in response to the comments, but they do NOT describe anything regarding this issue in the revised manuscript. As readers can only read what is written in the article, we should conclude that author can NOT justify this issue in this manuscript.

Second, as mentioned in the previous review, authors used receiver operating characteristics (ROC) curve to identify the threshold, but authors could not justify why threshold can be identified by ROC curve in method section. Authors suggested “This threshold allows us to discriminate between two populations (in this case amoxicillin-induced versus non-induced neurotoxicity) with acceptable sensitivity and specificity.”, but authors could NOT justify why authors concluded that “threshold allows us to discriminate” them and why authors concluded sensitivity and specificity were acceptable in the manuscript.

Finally, as mentioned in the previous review, authors used simple ROC without adjustment of potential confounding factors such as age, gender and GFR, but these potential confounding factors may affect the results. Authors suggest “we performed a multivariate logistic regression to assess the parameters”, but author continue to use ROC curve without adjustment of potential confounding factors for identifying threshold. If authors use logistic regression analysis for identification of threshold, authors should revies manuscript.

Minor comment

L75: “NS” should be spelled out.

L95: “sd” should be spelled out.

L175: “eGFR” should be spelled out.

Author Response

Manuscript Reference: antibiotics-1995807 intitled " Amoxicillin-induced neurotoxicity: contribution of a healthcare data
warehouse to the determination of a toxic concentration threshold"

Response to the referees

We would like to thank again the referees and the editor for their interesting new comments. We try to answer point by point to each of them.

Authors revised the manuscript, but this article has not fully answered some of the questions due to inadequate statistical analysis and insufficient description as mentioned in the previous review.

First, as mentioned in the previous review, authors excluded the patients who were not identified by “neurotoxicity query” from 2,054 patients who may receive amoxicillin, which may lead to selection bias. Authors claimed “We assume this bias because the initial query allows us to focus on an event with presumed low frequency. The exhaustive analysis of the 2054 medical records is not permitted by the data warehouse and the analysis of each medical record can only be performed after obtaining the query "neurotoxicity".” in response to the comments, but they do NOT describe anything regarding this issue in the revised manuscript. As readers can only read what is written in the article, we should conclude that author can NOT justify this issue in this manuscript.

Response : We really want to thank you for your sagacity. Indeed, this is an important point to discuss and it is undoubtedly a limitation of our work. We have added a sentence to highlight this issue in the discussion section with an objective of full transparency towards the reader. (L128-129)

Second, as mentioned in the previous review, authors used receiver operating characteristics (ROC) curve to identify the threshold, but authors could not justify why threshold can be identified by ROC curve in method section. Authors suggested “This threshold allows us to discriminate between two populations (in this case amoxicillin-induced versus non-induced neurotoxicity) with acceptable sensitivity and specificity.”, but authors could NOT justify why authors concluded that “threshold allows us to discriminate” them and why authors concluded sensitivity and specificity were acceptable in the manuscript.

Response : Indeed, we should have justified this threshold, you are right to underline again this point. To identify this threshold, we chose the Css value corresponding to the optimal specificity (86%). Indeed this choice is more relevant toward the clinical issue of amoxicillin-treated patients. A sentence dealing with this point was added (L116)

Finally, as mentioned in the previous review, authors used simple ROC without adjustment of potential confounding factors such as age, gender and GFR, but these potential confounding factors may affect the results. Authors suggest “we performed a multivariate logistic regression to assess the parameters”, but author continue to use ROC curve without adjustment of potential confounding factors for identifying threshold. If authors use logistic regression analysis for identification of threshold, authors should revies manuscript.

Response :  Sorry for the ambiguity, we should have made this more explicit. Indeed, the sole factor, which significantly influenced AMX induced neurotoxicity in the multivariate analysis, was AMX Css. (L75). Nonetheless, we performed a new ROC curve by forcing into the model all the other potential confounding factors: age, sex, weight, AMX daily dose, infection type, and eGFR. The identified threshold was identical to the threshold identified without adjustment of these confounding factors (109,7 vs 110 mg/L). We added complementary explanations in the method and result sections (L 74, 78-79 and 161-162).

Minor comment

L75: “NS” should be spelled out.

L95: “sd” should be spelled out.

L175: “eGFR” should be spelled out.

The corrections have been done

Round 3

Reviewer 1 Report

Authors revised the manuscript, but this article has not fully answered some of the questions due to inadequate statistical analysis and insufficient description.

First, authors used receiver operating characteristics (ROC) curve to identify the threshold, assuming “a maximal specificity target to identify this threshold to best response to the outcome (L129), but method and result are inconsistent. If a maximal specificity is threshold, the threshold will be the value at which specificity is 100% as shown in figure 3. Authors should carefully use statistical techniques.

Second, authors suggest “The AMX plasma concentrations threshold values were compared using the ROC curve build with and without adjustment of potential confounding factors”, but the detail of statistical method for ROC curve with adjustment as well as its reference is not shown in manuscript. It is difficult to judge the manuscript without the details.

Author Response

Manuscript Reference: antibiotics-1995807 intitled " Amoxicillin-induced neurotoxicity: contribution of a healthcare data
warehouse to the determination of a toxic concentration threshold"

Response to the referees

We would like to thank again the referees and the editor for their interesting new comments. We try to answer point by point to each of them.

Authors revised the manuscript, but this article has not fully answered some of the questions due to inadequate statistical analysis and insufficient description.

First, authors used receiver operating characteristics (ROC) curve to identify the threshold, assuming “a maximal specificity target to identify this threshold to best response to the outcome (L129), but method and result are inconsistent. If a maximal specificity is threshold, the threshold will be the value at which specificity is 100% as shown in figure 3. Authors should carefully use statistical techniques.

Response: Thank you again for you perspicacity. We have reviewed this point with our biostatistician. There is a misunderstanding from our part, the identification of this threshold for optimal specificity and acceptable sensitivity is based on the calculation of the Youden index [1]. This point was added in the material and method section (L166), in the results section (L76) and discussion section (L121).

[1]Youden WJ. Index for rating diagnostic tests. Cancer. 1950 Jan;3(1):32-5. 

Second, authors suggest “The AMX plasma concentrations threshold values were compared using the ROC curve build with and without adjustment of potential confounding factors”, but the detail of statistical method for ROC curve with adjustment as well as its reference is not shown in manuscript. It is difficult to judge the manuscript without the details.

Response : Indeed, this point is of importance to clearly understand the results. A sentence dealing with this point was added in the discussion (L122), material and methods section (L168) and the ROC curve with adjustment was added in supplementary material appendix (L81).

Round 4

Reviewer 1 Report

Authors revised the manuscript, but this article has not fully answered some of the questions due to insufficient description.

First, authors suggest “the multivariate analysis identified the same threshold value” (L80), but the detail of statistical result for ROC curve adjusted for potential confounding factors is not shown in main manuscript (only in supplement material). However, as authors show this result as one of main findings, it is also important to show detail of the result in main manuscript. Authors should add details of result regarding ROC curves adjusted for potential confounding factors in main manuscript.

Second, authors suggest “specificity superior to 80% and acceptable sensitivity” (L121), but they do not justify why “specificity superior to 80%” is meaningful and why its sensitivity is “acceptable” using references or other evidence. Authors should add description regarding this judgement with references in method section.

Author Response

Manuscript Reference: antibiotics-1995807 intitled " Amoxicillin-induced neurotoxicity: contribution of a healthcare data
warehouse to the determination of a toxic concentration threshold"

Response to the referees

We would like to thank again the referees and the editor for their supplementary comments. We try to answer point by point to each of them.

Authors revised the manuscript, but this article has not fully answered some of the questions due to insufficient description.

First, authors suggest “the multivariate analysis identified the same threshold value” (L80), but the detail of statistical result for ROC curve adjusted for potential confounding factors is not shown in main manuscript (only in supplement material). However, as authors show this result as one of main findings, it is also important to show detail of the result in main manuscript. Authors should add details of result regarding ROC curves adjusted for potential confounding factors in main manuscript.

Response: Thank you for this comment, we added these informations in the main manuscript.

Second, authors suggest “specificity superior to 80% and acceptable sensitivity” (L121), but they do not justify why “specificity superior to 80%” is meaningful and why its sensitivity is “acceptable” using references or other evidence. Authors should add description regarding this judgement with references in method section.

Response: We suppressed the qualification of sensitivity and specificity in the manuscript (L 124-125 suppressed).